# The impact of climate suitability, urbanisation, and connectivity on the expansion of dengue in 21st century Brazil

**Sophie A. Lee**[1,2]*, **Theodoros Economou**[3], **Rafael de Castro Catão**[4], **Christovam Barcellos**[5], **Rachel Lowe**[1,2,6]

**1** Centre for Mathematical Modelling of Infectious Diseases, London School of Hygiene & Tropical Medicine, London, United Kingdom, **2** Centre on Climate Change and Planetary Health, London School of Hygiene & Tropical Medicine, London, United Kingdom, **3** Climate and Atmosphere Research Centre, The Cyprus Institute, Nicosia, Cyprus, **4** Departamento de Geografia, Universidade Federal do Espírito Santo, Vitoria, Brazil, **5** Fundação Oswaldo Cruz, Rio de Janeiro, Brazil, **6** Barcelona Supercomputing Center, Barcelona, Spain

* sophie.lee@lshtm.ac.uk

**Data Availability Statement:** All data used in this study is open access and available freely on the internet, see the methods section for more details.

## Abstract

Dengue is hyperendemic in Brazil, with outbreaks affecting all regions. Previous studies identified geographical barriers to dengue transmission in Brazil, beyond which certain areas, such as South Brazil and the Amazon rainforest, were relatively protected from outbreaks. Recent data shows these barriers are being eroded. In this study, we explore the drivers of this expansion and identify the current limits to the dengue transmission zone. We used a spatio-temporal additive model to explore the associations between dengue outbreaks and temperature suitability, urbanisation, and connectivity to the Brazilian urban network. The model was applied to a binary outbreak indicator, assuming the official threshold value of 300 cases per 100,000 residents, for Brazil's municipalities between 2001 and 2020. We found a nonlinear relationship between higher levels of connectivity to the Brazilian urban network and the odds of an outbreak, with lower odds in metropoles compared to regional capitals. The number of months per year with suitable temperature conditions for *Aedes* mosquitoes was positively associated with the dengue outbreak occurrence. Temperature suitability explained most interannual and spatial variation in South Brazil, confirming this geographical barrier is influenced by lower seasonal temperatures. Municipalities that had experienced an outbreak previously had double the odds of subsequent outbreaks. We identified geographical barriers to dengue transmission in South Brazil, western Amazon, and along the northern coast of Brazil. Although a southern barrier still exists, it has shifted south, and the Amazon no longer has a clear boundary. Few areas of Brazil remain protected from dengue outbreaks. Communities living on the edge of previous barriers are particularly susceptible to future outbreaks as they lack immunity. Control strategies should target regions at risk of future outbreaks as well as those currently within the dengue transmission zone.

Data and code used to produce this analysis is available from a Github repository cited in the manuscript (https://github.com/sophie-a-lee/Dengue_expansion).

**Funding:** S.A.L. was supported by a Royal Society Research Grant for Research Fellows. https://royalsociety.org/. R.L. was supported by a Royal Society Dorothy Hodgkin Fellowship https://royalsociety.org/. T.E. was funded by the European Union's Horizon 2020 research and innovation programme under grant agreement No. 856612 https://ec.europa.eu/info/research-and-innovation/funding/funding-opportunities/funding-programmes-and-open-calls/horizon-europe_en and the Cyprus Government. C.B. was funded by the National Council for Scientific and Technological Development under grant No. 303985/2019-4 (http://www.cnpq.br/) The funders had no role in study design, data collection and analysis, decision to publish, or preparation of the manuscript.

**Competing interests:** The authors have declared that no competing interests exist.

## Author summary

Dengue is a mosquito-borne disease that has expanded rapidly around the world due to increased urbanisation, global mobility and climate change. In Brazil, geographical barriers to dengue transmission exist, beyond which certain areas including South Brazil and the Amazon rainforest are relatively protected from outbreaks. However, we found that the previous barrier in South Brazil has shifted further south as a result of increased temperature suitability. The previously identified barrier protecting the western Amazon no longer exists. This is particularly concerning as we found dengue outbreaks tend to become established in areas after introduction. Highly influential cities with many transport links had increased odds of an outbreak. However, the most influential cities had lower odds of an outbreak than cities connected regionally. This study highlights the importance of monitoring the expansion of dengue outbreaks and designing disease prevention strategies for areas at risk of future outbreaks as well as areas in the established dengue transmission zone.

## Introduction

Dengue is considered one of the top 10 threats to global health [1], with around half the world's population living in areas at risk of infection [2]. Incidence rates have doubled each decade in the past 30 years as a result of increased urbanisation, global mobility and climate change [2–4]. All 4 dengue serotypes are endemic to Brazil, which experiences frequent outbreaks across the country [5]. Previous studies identified geographical barriers to dengue transmission beyond which regions were relatively protected. This included South Brazil, where seasonal temperatures are too cold for vectors to efficiently transmit the virus, areas of high altitude in Southeast Brazil and remote regions of the western Amazon [6]. However, these barriers are being eroded and the dengue transmission area in Brazil has expanded over the past decade. This expansion is thought to be linked to increased human mobility and changes in climate [7,8].

For dengue to become established in a new region, the environment must be suitable to support the propagation of the dengue vector, *Aedes* mosquitoes. There are two vectors present in Brazil capable of transmitting the dengue virus: *Aedes aegypti* and *Aedes albopictus*. Currently only *Aedes aegypti* are considered responsible for dengue transmission in Brazil [9,10], however a recent study identified *Aedes albopictus* infected by dengue virus in a rural area of Brazil during an outbreak, which could indicate their involvement in the introduction of dengue to rural areas [11]. *Aedes aegypti* have evolved to live in urban environments close to humans [12] but there is evidence to suggest they are becoming established in peri-urban and rural regions of South America [13,14]. Conversely, *Aedes albopictus* are typically found in peri-urban areas but have been identified in densely urbanised areas such as urban slums in Brazil [9,15]. *Aedes* mosquitoes breed in pools of standing, clean water created by water storage containers or uncollected refuse. These conditions arise when rapid urbanisation occurs without adequate improvements to infrastructure, such as access to piped water and refuse collection [16,17]. There is evidence that areas lacking reliable access to piped water are more susceptible to dengue outbreaks, particularly in highly urbanised areas following drought [18]. Prior studies have found that extremely wet conditions also increased the risk of dengue outbreaks, thought to be linked to the creation of larval habitat in the short term [18,19]. Suitable temperature conditions are required for the mosquitoes to breed and transmit the virus. *Aedes aegypti* are unable to survive in temperatures below 10˚C or above 40˚C [20] and can only

transmit the virus between 17.8˚C and 34.5˚C [21,22]. *Aedes albopictus* are more suited to cooler temperatures and can transmit the virus between 16.2˚C and 31.4˚C [21,22]. Recent outbreaks in temperate cities of South America have shown that epidemics are still possible in regions that experience seasonal temperatures outside of this range due to human movement [23–25].

The expansion of *Aedes aegypti* and the arboviruses they transmit into rural parts of the Amazon has been linked to connections to and within the area by air, road or boat [13,26]. Despite this, the investigation of spatial connections created by human movement is little explored in the literature and the vast majority of spatial modelling studies of mosquito-borne diseases assume connectivity is based on distance alone [27]. Brazilian cities are connected to one another within a complex urban network, described within the Regions of influence of cities ("Regiões de Influência das Cidades", REGIC) studies carried out by the Brazilian Institute of Geography and Statistics [28,29]. People often travel great distances to reach large urban centres as they contain important educational, business or cultural institutions. Failure to account for long-distance movements may miss important drivers of dengue expansion, particularly in areas such as the Amazon where the average distance travelled to Manaus, the capital of Amazonas state, was 316km. Important cities can have influence over vast areas of Brazil, for example the region of influence connected to the capital city of Brasilia corresponds to over 20% of the country and spans 1.8 million km$^2$ [29].

Although previous studies have shown the expansion of dengue outbreaks in Brazil [7] and the association between the dengue transmission zone and climate [6], neither formally investigated the link between this expansion and human movement. In this study, we use the level of influence of cities from the REGIC studies [28,29] as a proxy for human movement, and aim to better understand how climate suitability, connectivity between cities and socioeconomic factors have contributed to the recent expansion of dengue. It is hoped that by understanding the drivers of dengue expansion in Brazil, we can identify its spatial trends and regions at risk from future outbreaks.

## Methods

### Epidemiological data

Brazil is the 6th most populous country in the world with an estimated population of over 212 million in 2020 [30]. The country can be separated into 5 distinct geo-political regions (S1 Fig), 27 federal units (26 states and a federal district containing the capital city Brasilia, S1 Fig), and 5,570 municipalities. We obtained monthly notified dengue cases for each of Brazil's 5,570 municipalities between January 2001 and December 2020 from Brazil's Notifiable Diseases Information System (SINAN), freely available via the Health Information Department, DATASUS (https://datasus.saude.gov.br/informacoes-de-saude-tabnet/). Cases were aggregated by month of first symptom and municipality of residence. Dengue cases are considered confirmed if they test positive in a laboratory or, more commonly, based on the Ministry of Health's syndromic definition Due to its passive nature, the accuracy of the dengue surveillance system differs between municipalities and between periods of high and low incidence [31]. To reduce the bias introduced by differences in case reporting and health seeking behaviour, we chose to model binary outbreak indicators rather than incidence rates because, as stated by the Brazilian Observatory of Climate and Health, "there is no way to hide an epidemic" [6]. Between 2001 and 2020, municipality boundaries in Brazil have changed and several new municipalities were created. To ensure data were consistent over the study period, we aggregated data to the 5,560 municipalities that were present in 2001 by combining the new municipalities with their parent municipalities. The data and code used to aggregate the dengue case data are available from https://github.com/sophie-a-lee/Dengue_expansion [32].

## Meteorological data

Monthly mean temperatures (K) were obtained from the European Centre for Medium-Range Weather Forecasts' (ECMRWF) ERA5-Land dataset [33] for the period January 2001—December 2020, at a spatial resolution of 0.1˚ x 0.1˚ (~9km). The ERA5-Land database was chosen because of its fine spatial scale, necessary when analysing small administrative units such as municipalities. Temperatures were converted from Kelvin to degrees Celsius (˚C) by subtracting 273.15. Mean temperature was aggregated to each municipality using the exactextractr package [34] in R (version 4.0.3) by calculating the mean of the grid boxes lying within each municipality. Grid boxes partially covered by a municipality were weighted by the percentage of area that lay within the municipality.

Due to its size, Brazil experiences a wide range of climate systems and ecosystems. The northern part of the country lies on or close to the equator, meaning regions experience year-round high temperatures. In contrast, the South and Southeast regions have clear seasonality in temperatures with cooler winters (S2 Fig), often falling below the optimal temperature range for dengue transmission (between 17.8˚C and 34.5˚C for *Aedes aegypti* and 16.2˚C and 31.4˚C for *Aedes albopictus* [21,22]). To understand how temperature suitability has contributed to the expansion of the dengue transmission zone in Brazil, we calculated the number of months per year each municipality lay within the suitable temperature ranges (between 16.2˚C and 34.5˚C). Most of Brazil experiences year-round temperature suitability except for the temperate South and mountainous regions in the Southeast (S3 Fig), although the number of months suitable has increased in these regions over the past decade (Fig 1). As *Aedes aegypti* is the only vector proven to transmit dengue in Brazil, we also tested the number of months considered suitable for *Aedes aegypti* transmission (between 17.8˚C and 34.5˚C) within the model.

## Urbanisation

We obtained the percentage of residents in each municipality living in urban areas from the 2000 and 2010 censuses via DATASUS. In 2010, just under 85% of Brazil's population lived in urban areas, mostly concentrated in the large cities of South and Southeast Brazil. The North region, except for some state capitals, has a larger rural population (S4 Fig). The percentage of residents living in urban areas was converted to the proportion to make interpretation and comparison of model coefficients easier. Data from the 2000 census was used for the years 2001–2009 and data from 2010 was used for the years 2010–2020 to account for changes in urbanisation over the period. Further details on the socioeconomic variables considered in this analysis are given in S1 Text.

## Hierarchical levels of influence of cities

As a proxy for human movement, we obtained the hierarchical level of influence of cities from IBGE's REGIC studies, carried out in 2007 and 2018 [28,29]. REGIC aims to recreate the complex urban network of Brazil using information from surveys about the frequency and reasons for the movement of people and goods around the country. Part of this study involved classifying cities based on their hierarchical level of influence within this network (see S1 Text for more details). Cities were classified into five levels:

1. Metropolis: the largest cities in Brazil, with strong connections throughout the entire country. This includes São Paulo, the capital Brasilia, and Rio de Janeiro.

2. Regional capital: large cities which are connected throughout the region in which they are located and to metropoles. This includes state capitals that were not classified as metropoles, such as Rio Branco, Campo Grande and Porto Velho.

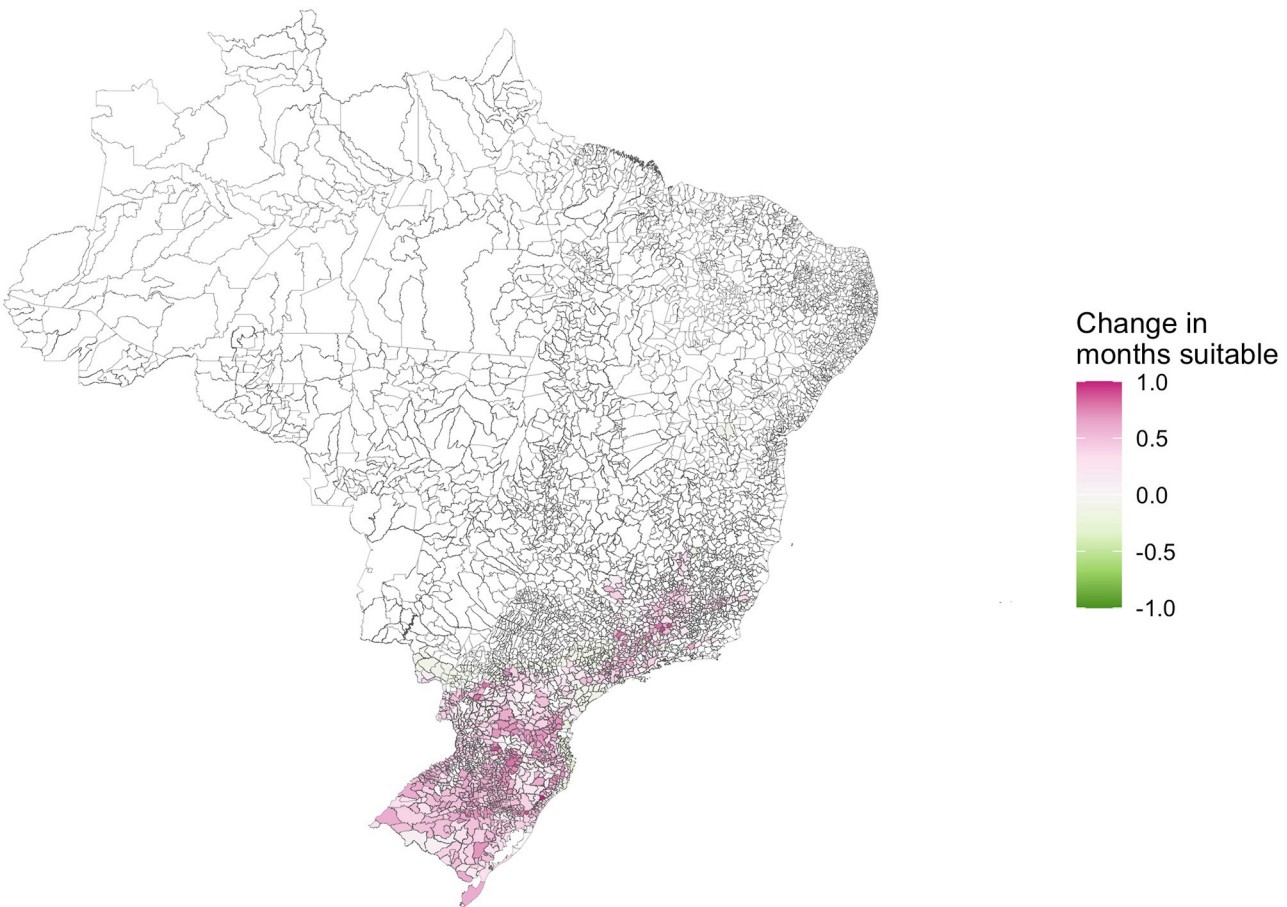

**Fig 1. The difference between the average number of months with suitable temperatures for dengue transmission in 2001–2010 and 2011–2020.**
The number of months with temperatures between 16.2˚C and 34.5˚C has increased on average (shown in pink) in parts of South and Southeast Brazil which were previously considered 'protected' from dengue transmission. Maps were produced in R using the geobr package [32,35] (https://ipeagit. github.io/geobr/).

3. Sub-regional capital: cities with a lower level of connectivity, mostly connected locally and to the three largest metropoles.

4. Zone centre: smaller cities with influences restricted to their immediate area, often neighbours.

5. Local centre: the smallest cities in the network which typically only serve residents of the municipality and are not connected elsewhere.

The REGIC study aggregated data to population concentration areas ("Áreas de Concentração de População", ACPs), defined in [36]. Smaller or isolated ACPs consisted of a single municipality, while large urban centres consisted of multiple municipalities. Levels of influence were extracted for each municipality based on the ACP they belonged to, meaning small municipalities neighbouring large cities may have a high level of influence. The distribution of highly connected urban centres is uneven across the country; the South and Southeast regions are particularly well connected, while the North and Northeast contain fewer high-level centres (Fig 2 and S1 Table). To account for any changes in connectivity over the study period, we

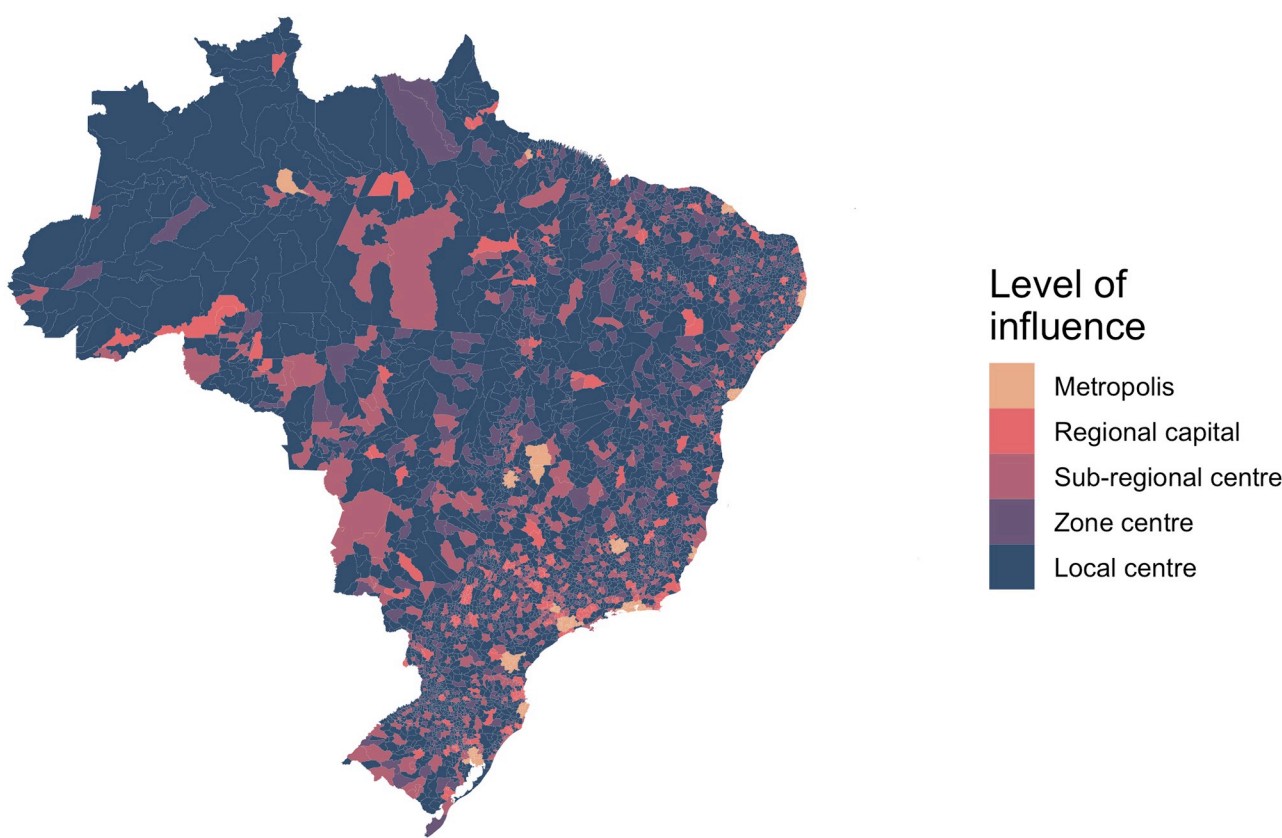

**Fig 2. The level of influence of cities within the Brazilian urban network from REGIC 2018.** The Amazon region is far less connected to the urban network than the rest of the country. As there is only one metropolis in North Brazil (Manaus), people often travel great distances, far greater than in other regions, to reach cities. Maps were produced in R using the geobr package [32,35] (https://ipeagit.github.io/geobr/).

used the levels extracted from the 2007 study for the years 2001–2010, and levels from the 2018 study for the years 2011–2020.

## Modelling approach

To understand how the dengue transmission zone has expanded between 2001 and 2020, we aggregated dengue cases by year and created a binary outbreak indicator. We used an outbreak threshold of more than 300 cases per 100,000 residents, defined as 'high risk' by the Brazilian Ministry of Health [37]. We also tested a 'medium risk' indicator, defined as more than 100 cases per 100,000 residents, and a threshold defined as the 75th percentile of the dengue incidence rate between 2001–2020 for each municipality to ensure our analyses were robust to this outbreak definition. The annual dengue incidence rate was calculated using estimates of the annual population for each municipality obtained from the Brazilian Institute of Statistics and Geography (IBGE) via DATASUS (http://tabnet.datasus.gov.br/cgi/deftohtm.exe?ibge/cnv/poptbr.def). Further details about the dengue surveillance system in Brazil and outbreak definitions are given in S1 Text. We formulated a binomial spatio-temporal generalised additive model (GAM) using the binary outbreak indicator as the response variable. We included the number of months per year with temperature suitable for *Aedes* mosquitoes to transmit dengue, the level of influence from REGIC, the proportion of residents living in urban areas, and a 'prior outbreak' indicator which took the value 0 until the year of the first outbreak in a municipality and 1 in every year after as covariates. We also considered the number of extremely wet

months as a covariate, but we found this did not improve the model (further details can be found in S1 Text). To account for spatial and temporal patterns in the data, smooth functions of the year and the coordinates of the centroids of municipalities were included in the model (see S1 Text for further details). Inference was performed using an empirical Bayesian approach with estimates calculated using restricted maximum likelihood (REML) as part of the mgcv package in R [38].

Model fit was assessed using a receiver operating characteristic (ROC) curve which plots the true positive rate against the true negative rate at different thresholds to test the predictive ability of the model. The area under the ROC curve was calculated as this gives a measure of predictive ability compared to chance, which would return a value of 0.5. The closer the area under the ROC curve is to 1, the better the model fits the data. The predictive ability of models were also compared using the Brier score [39]. The Brier score is the mean squared difference between the observed and expected outcomes; a lower Brier score represents a better fitting model.

To assess the relative contribution of the covariates, we compared the spatio-temporal structured residual terms between the final model and a baseline model, containing only the spatio-temporal smooth terms. If the covariates explained variation in the data, the smooth functions would shrink towards zero in the final model and the difference between the absolute estimates of these functions would be negative. To assess the contribution of the covariates over the entire period, we took the median difference for each municipality. The contribution of each individual covariate was also assessed by taking the difference between the structured residuals from the baseline model and models with each covariate added in turn.

To understand how the risk of outbreaks have changed between 2001 and 2020, we drew 1000 simulations from the posterior distribution of the response and estimated the probability of an outbreak for each municipality per year. These estimates were aggregated to the first (2001–2010) and second (2011–2020) decades by taking the mean probability for each municipality per decade to observe how the dengue transmission zone had changed after the large-scale outbreak of the 21st century in 2010. The estimated probabilities were then used to determine the current dengue transmission barriers by identifying regions where the average probability of an outbreak lay below 10%, other barrier thresholds were also considered.

## Results

There were 13,860,348 cases of dengue notified between January 2001 and December 2020 in Brazil. The dengue incidence rate has increased across all regions of the country (Fig 3) particularly in the Centre-West and Southeast. Outbreaks were more widespread since 2010 with around 80% of all municipalities in the Centre-West now regularly experiencing outbreaks (S5 Fig). Although the South had the highest incidence in 2020, this was still concentrated in a small number of municipalities in Paraná, around the fringe area of the previously identified geographical barrier. The previous barriers to dengue transmission have been eroded over the past decade. This is particularly noticeable in the western Amazon where there are now very few municipalities yet to experience an outbreak. The erosion of the barrier in the South was particularly noticeable in 2020 when it had the highest incidence rate of any region (Fig 3), and many municipalities close to the previous barrier experienced outbreaks for the first time (Fig 4). We observed that once dengue was introduced to municipalities, the virus became established and future outbreaks were likely to occur (Fig 5).

### Model results

We found municipalities that were highly urbanised, highly connected, and had temperatures suitable for dengue transmission year-round had a significantly increased odds of an outbreak

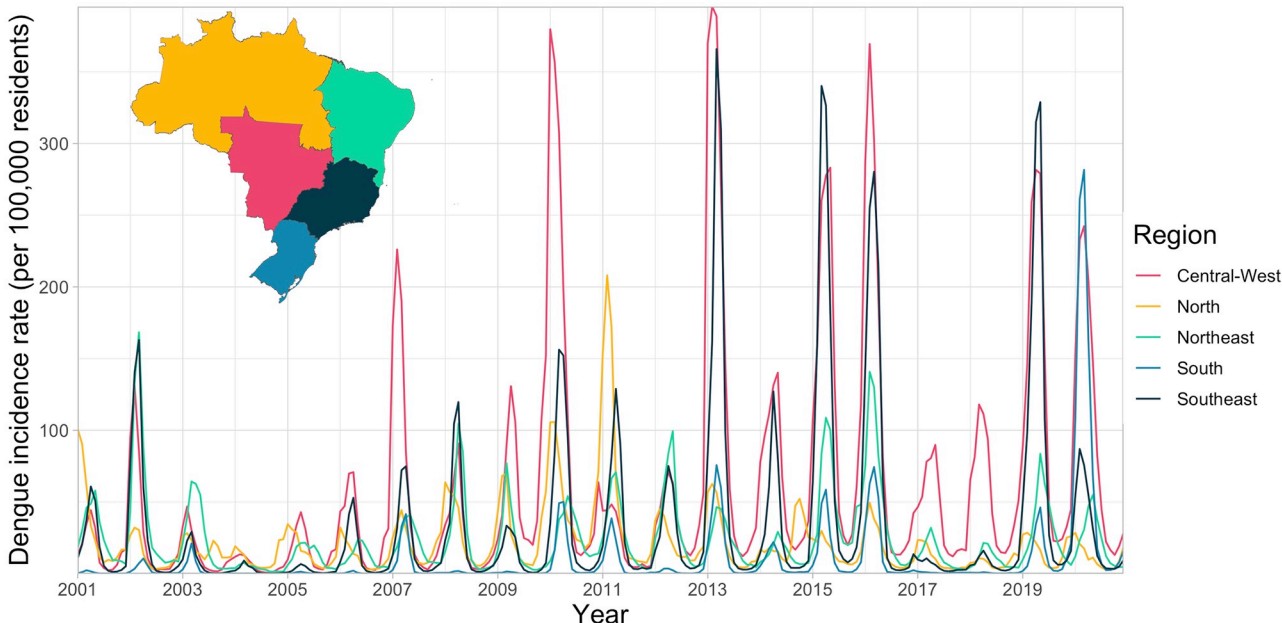

**Fig 3. Monthly incidence rate per 100,000 residents in regions of Brazil 2001–2020.** Incidence rates have increased in every region of the country between 2001–2020. The first regional outbreak occurred in 2010, outbreaks have occurred more frequently and in more regions since then. Maps were produced in R using the geobr package [32,35] (https://ipeagit.github.io/geobr/).

(Table 1 and Fig 6). Municipalities that had previously experienced outbreaks had around double the odds of experiencing another compared to municipalities that were still protected (adjusted odds ratio (aOR): 2.03, 95% credible interval (CI): 1.93, 2.15). This could indicate that the virus becomes established following its introduction, however the increased incidence may be a result of increased surveillance following an outbreak or due to the increased probability of severe cases following the introduction of new serotypes [40]. Municipalities with

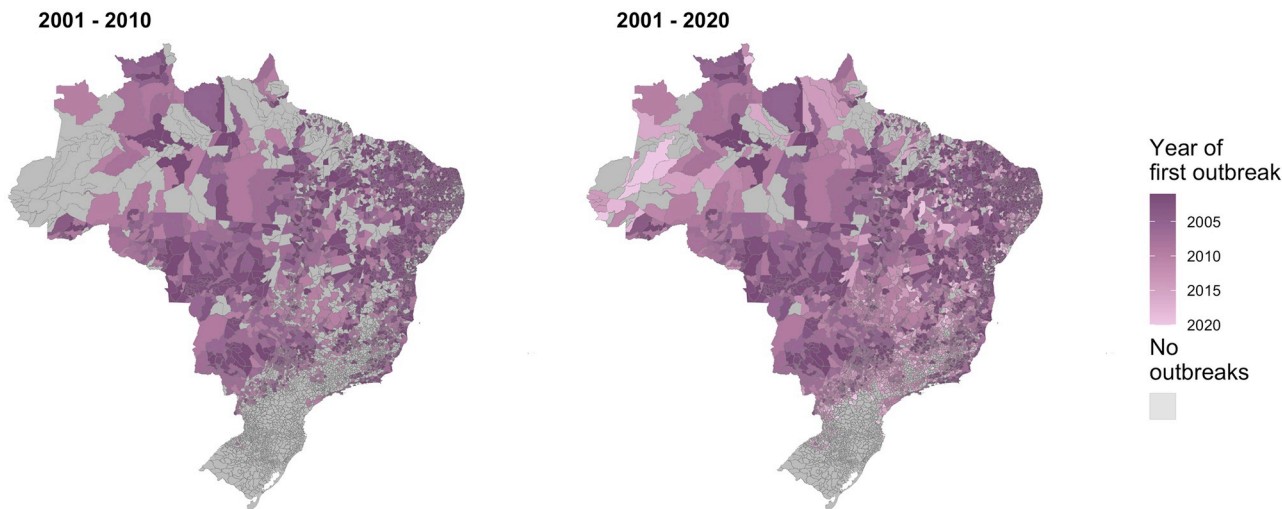

**Fig 4. The first year each municipality experienced an outbreak for the first time in the period 2001–2010 and 2001–2020.** The year each municipality first recorded over 300 cases per 100,000 residents. Recent data shows the previous barriers to dengue outbreaks in the Amazon and South are being eroded. Maps were produced in R using the geobr package [32,35] (https://ipeagit.github.io/geobr/).

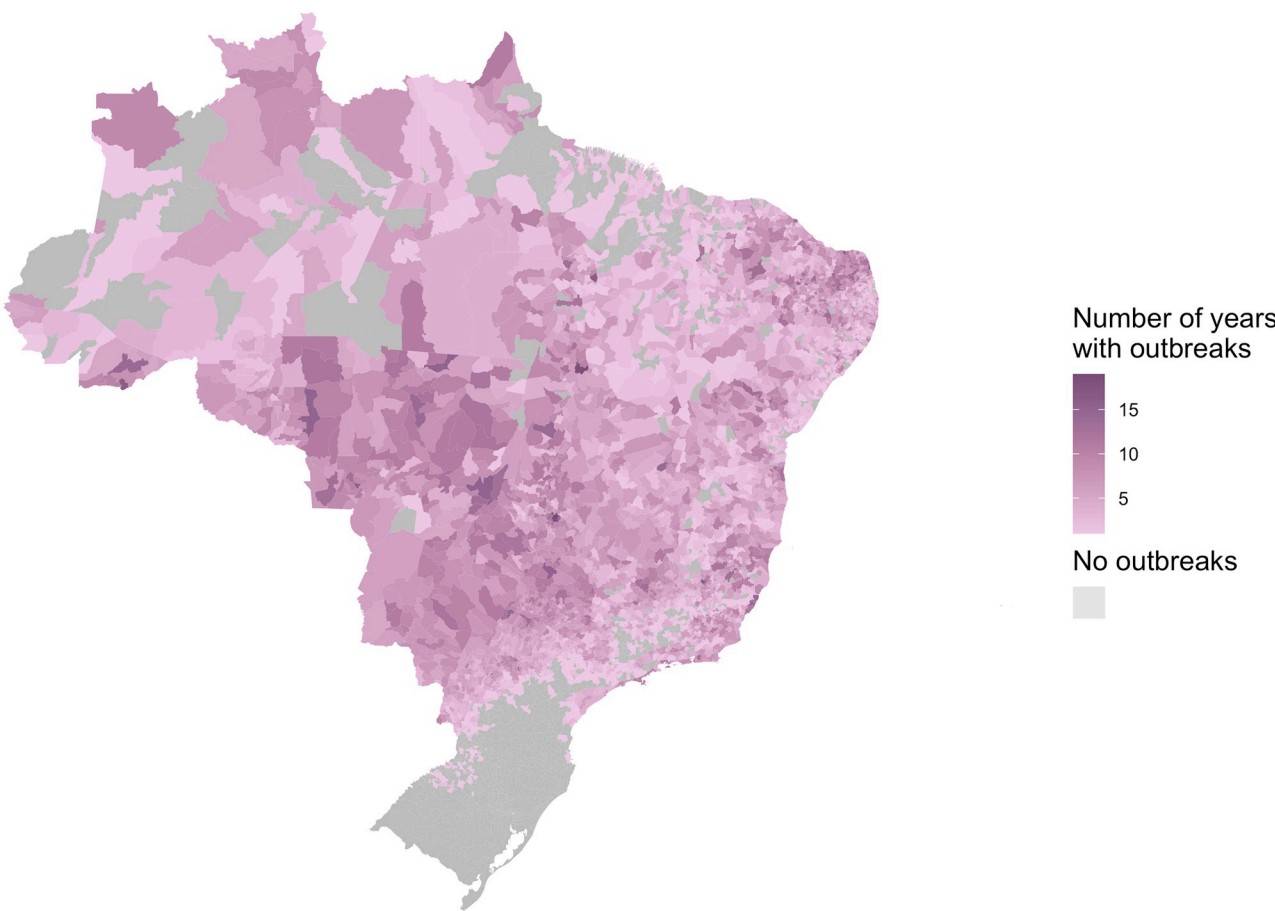

**Fig 5. The number of years each municipality experienced an outbreak between 2001 and 2020.** Municipalities that experienced outbreaks earlier in the 21st century continued to experience outbreaks throughout the period. This suggests that once dengue is introduced to a region, it becomes established. Maps were produced in R using the geobr package [32,35] (https://ipeagit.github.io/geobr/).

**Table 1. Posterior mean and 95% credible interval (CI) estimates for linear effect parameters, shown on the adjusted odds ratio (aOR) scale.**

| Coefficient | aOR (95% CI) |
|---|---|
| Urbanisation | 3.26 (2.85, 3.72) |
| REGIC level: metropolis | 1.39 (1.22, 1.59) |
| REGIC level: regional capital | 1.52 (1.38, 1.66) |
| REGIC level: sub-regional centre | 1.23 (1.14, 1.33) |
| REGIC level: zone centre | 1.23 (1.15, 1.31) |
| Prior outbreak: yes | 2.03 (1.93, 2.15) |
| Months with suitable temperature | 1.42 (1.30, 1.55) |

Posterior mean and credible interval estimated taking the 50th, 2.5th and 97.5th quantiles from the simulated posterior distribution. The response variable is a dengue outbreak, defined as over 300 cases per 100,000 residents. Urbanisation is the proportion of residents living in urban areas. REGIC covariates are in comparison to the reference group, local centre. A suitable temperature is defined as between 16.2˚C and 34.5˚C (suitable for both *Aedes aegypti* and *Aedes albopictus*).

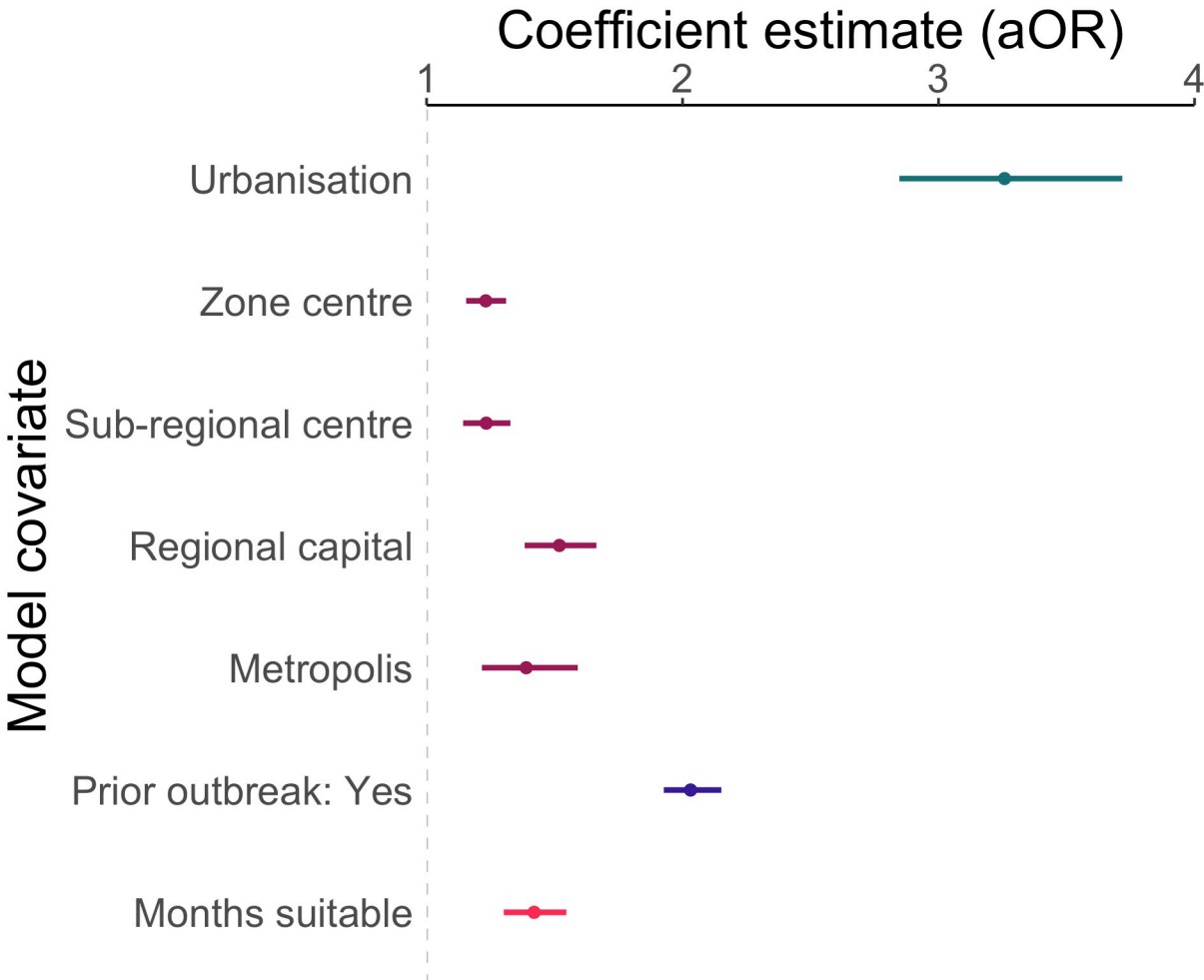

**Fig 6. The mean and 95% credible interval of the posterior distribution for each model covariate.** Results show that municipalities with a higher proportion of residents living in urban areas, in cities with a higher connectivity than local centres, with a higher number of month per year suitable for dengue transmission, which had previously experienced an outbreak have significantly higher odds of an outbreak.

year-round temperature suitability had increased risk of outbreaks, whether we consider suitability for both species of *Aedes* mosquitoes (Table 1) or just *Aedes aegypti* (S2 Table). On average, the odds of an outbreak increased by 42% (aOR: 1.42, 95% CI: 1.30, 1.55) for every additional month of suitable temperature per year.

Although higher levels of connectivity had significantly higher odds of an outbreak than local centres, this difference was highest on average for regional centres (aOR: 1.52, 95% CI: 1.38, 1.66) despite being considered less connected to the urban network than metropoles (aOR: 1.39, 95% CI: 1.22, 1.59). This is potentially due to the structure of the urban network which connects smaller cities to larger centres until they converge to metropoles, meaning that regional capitals are important intermediate urban centres, that influences wide hinterland areas [29]. Alternatively, despite the regional capitals having similar levels of access to basic services as metropoles when aggregated to the municipality level (S6 Fig), metropoles have larger economies and greater access to healthcare than regional capitals [29] which may mean improved infrastructure which is not reflected by census variables on this scale.

The area under the ROC curve for the final model was 0.86 (95% confidence interval: 0.856, 0.861, S7 Fig), indicating that the model fit the data well. We found the conclusions drawn from the models using alternative outbreak definitions remained consistent, however the coefficient values differed (S8 Fig and S2 Table). In particular, the model based on the 75th percentile produced lower coefficient estimates than the fixed threshold models, and the model using a threshold of over 100 cases per 100,000 residents estimated an increased odds following a previous outbreak compared to the primary analysis (S8 Fig and S2 Table). We found that the fixed threshold models outperformed the 75th percentile according to the ROC curve (S7 Fig) and Brier score (S3 Table). The temporal smooth function showed increasing odds of an outbreak over the period not explained by the model covariates (Fig 7). The spatial smooth field showed that the risk around Rio Branco in Acre, the Centre-West region, and in Rio Grande do Norte in Northeast Brazil were higher on average than explained by the model covariates (Fig 7). In contrast, areas in South Brazil, along the northern Brazilian coast, and in parts of the Amazon had lower risk of dengue outbreak occurrence than expected given the covariates.

The structured residuals for the full model were closer to zero on average for the vast majority of the country than the baseline model (92.33% of municipalities, Fig 8), indicating that the covariates are indeed explaining spatio-temporal variation in the data. The inclusion of temperature suitability into the baseline model shrank the structured residuals towards zero for 91.16% of municipalities. This was particularly noticeable in South Brazil (Fig 9), supporting the hypothesis that the dengue transmission barrier here was a result of lower temperatures. The inclusion of the prior outbreak indicator also shrank the structured residuals towards zero across Brazil (in 94.28% of municipalities, Fig 9) showing its relative importance in this model. The relative importance of urbanisation and REGIC levels of influence were less clear; despite the model finding both these variables significantly associated with increased odds of an outbreak, there were fewer municipalities in which the structured residuals had shrank towards (57.5% for urbanisation, Fig 9, and 45.08% for REGIC levels of influence, Fig 9). One potential reason for this is that both variables are only measured once per decade and therefore do not

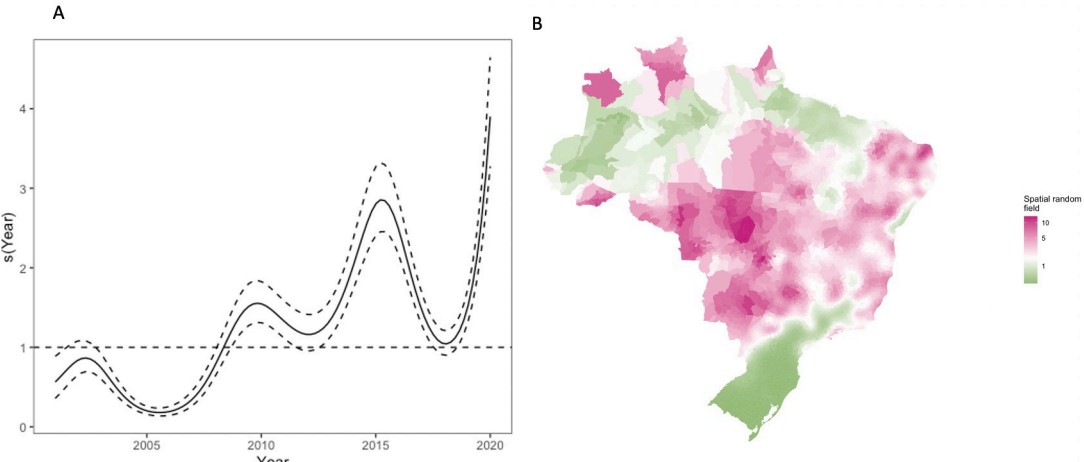

**Fig 7. Temporal (a) and spatial (b) smooth functions from the final model transformed to show the change in odds.** The odds of an outbreak has increased over the period due to unexplained factors not included in the model. The spatial random field highlights that more information is needed in the model to understand the explosive outbreaks that have taken place in Rio Branco, Acre and the Centre-West region as these hotspots are not fully explained by the model covariates. Pink (green) regions of the map represent areas where the odds of an outbreak was higher (lower) on average than estimated by the covariates. Maps were produced in R using the geobr package [32,35] (https://ipeagit.github.io/geobr/).

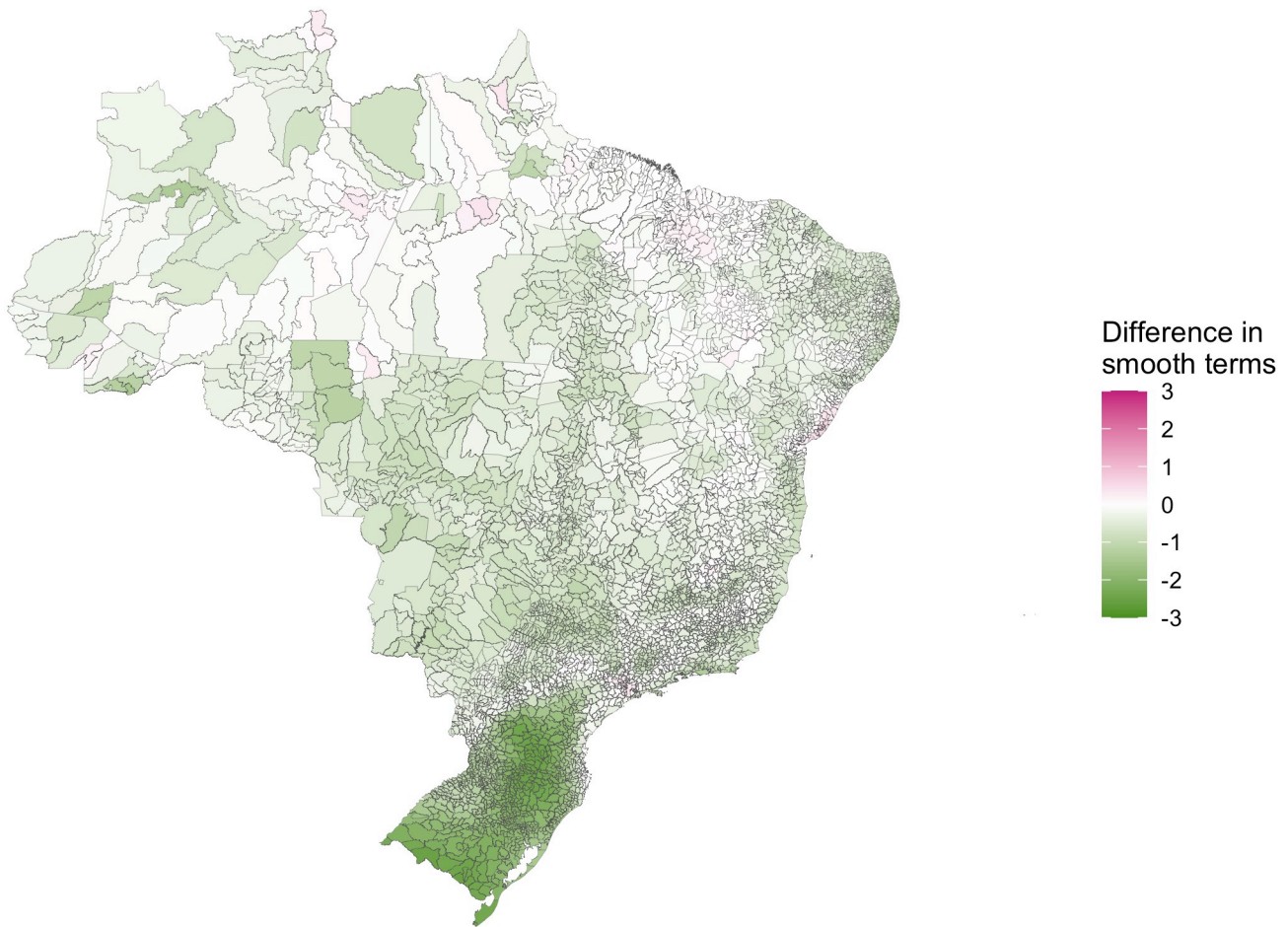

**Fig 8. The median difference between absolute values of the smooth function estimates calculated from the full model and from a baseline model.**
A reduction in the absolute smooth functions (shown in green) indicates that the estimates have shrunk towards zero when the covariates were added to the model and these covariates are explaining some of the variability in the data. Maps were produced in R using the geobr package [32,35] (https://ipeagit.github.io/geobr/).

differ annually; there may be changes in municipalities that contribute to dengue transmission but are not captured by these stationary variables. Another potential reason is that these variables are not able to account for within-city variation at this spatial resolution that may contribute to outbreaks of dengue.

The probability of an outbreak increased across most of Brazil since the first decade of the 21st century except for the 2 most southern states and some areas of the Northeast (Fig 10). The largest increases in risk were seen in the Centre-West, which has been the epicentre of the explosive outbreaks taking place since 2010. In the regions previously protected from outbreaks (the western Amazon and the South (Fig 10)), the erosion of the geographic barriers can clearly be seen. Although a southern border still exists, it has shifted south, and the Amazon no longer has a clear boundary.

## Current barriers to dengue transmission

To determine the current dengue transmission barriers, we identified regions where the average probability of an outbreak lay below 10% (Fig 11). We chose the threshold 10% as this gave

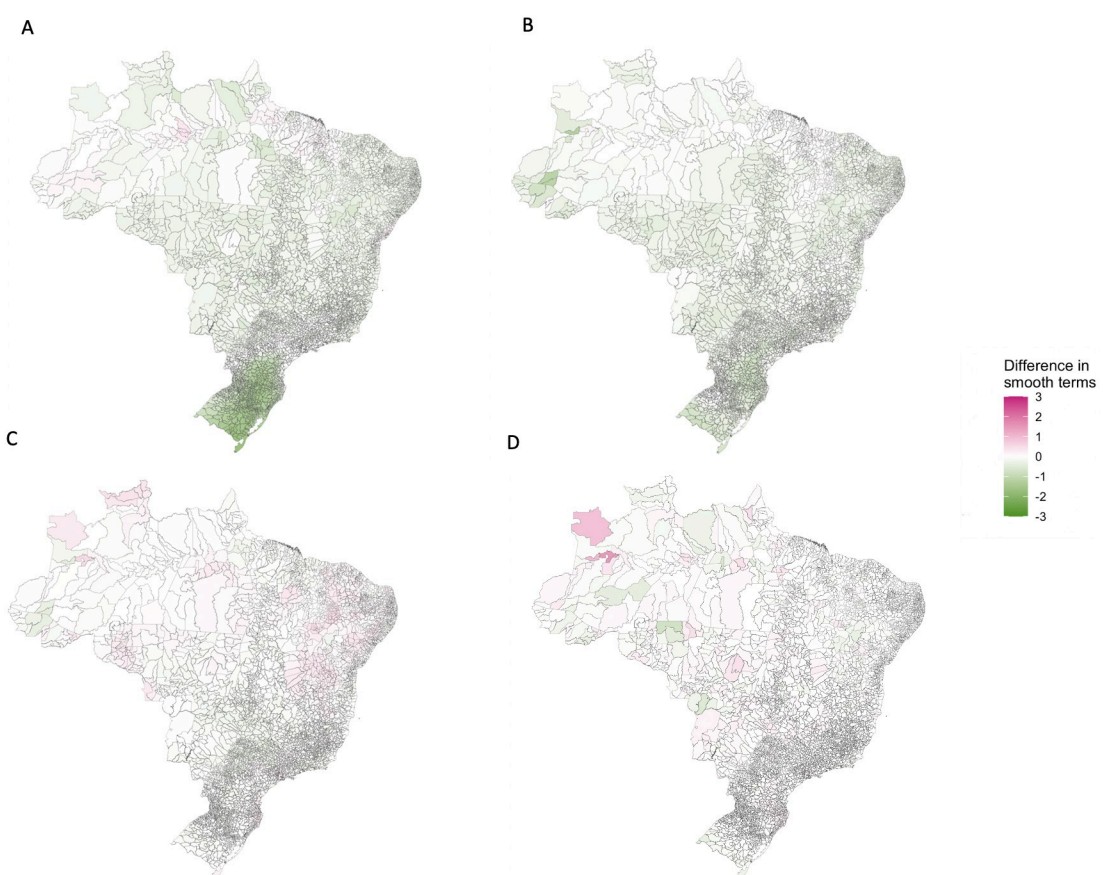

**Fig 9. The median difference between absolute values of the smooth function estimates calculated from the baseline model and models with a) the climate suitability covariate added, b) the prior outbreak indicator added, c) the proportion of urbanisation added, and d) the level of connectivity covariate added.** A reduction in the absolute estimates of the smooth functions (shown in green here) indicates that the functions have shrunk towards zero and the covariate has explained variation in the data. Maps were produced in R using the geobr package [32,35] (https://ipeagit.github.io/geobr/).

barriers comparable to those identified in a previous study [6] (S10 Fig). The number of municipalities considered protected declined from 2689 in 2001–2010 to 1599 in 2011–2020. Between 2011 and 2020 there were no municipalities in the Centre-West region that were considered protected, compared to 92 in 2001–2010. Northeast Brazil was the only region that had more protected municipalities in 2011–2020 than 2001–2020 (366 compared to 315). The southern barrier to dengue transmission now begins in the southern part of Paraná and extends through the west of Rio Grande do Sul and Santa Catarina. Areas of high altitude in Southeast Brazil, mostly found in Minas Gerais, are still considered protected. There are still areas of the Amazon protected from dengue outbreaks, but this barrier is no longer clearly defined. In addition to the previously identified barriers in the South region and Amazon rainforest, we found that there was a protected region along the north coast of Brazil in northern Pará and Maranhão. This barrier was not explained by the covariates in our model indicated by the low values of the spatial smooth function (Fig 7). This area is predominantly warm and humid climate, with higher precipitation during winter ('Am' type in Köppen climate classification) [41]. Although temperature and humidity are relatively stable along seasons in this area, the interaction between these variables and increased precipitation may inhibit the mosquito populations [42].

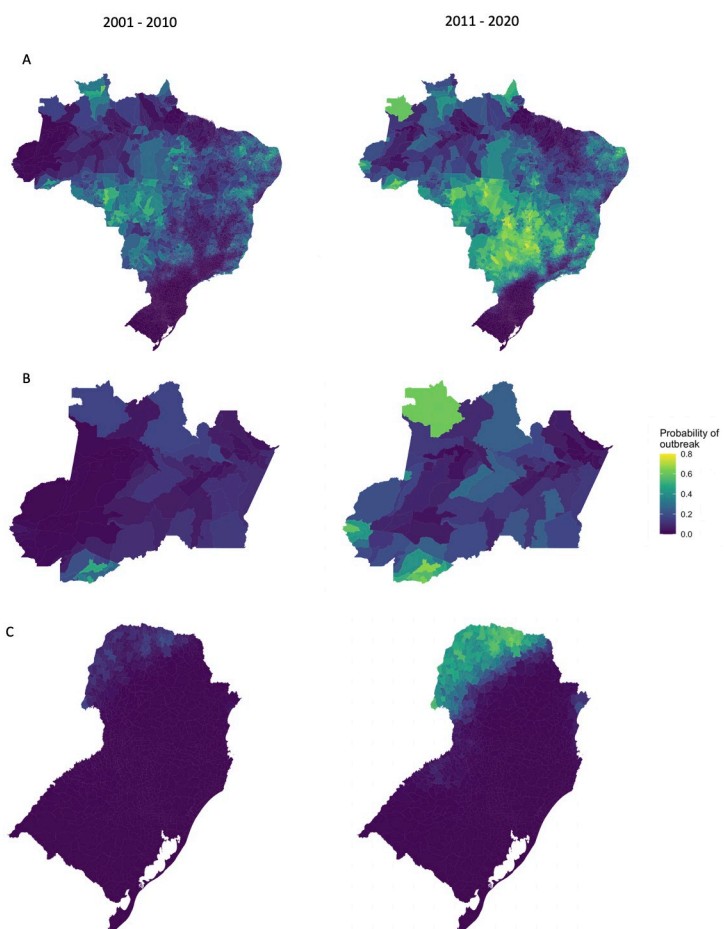

**Fig 10. The average probability of an outbreak 2001–2010 and 2011–2020 in a) Brazil, b) Acre and Amazonas, and c) South Brazil.** The probability of an outbreak estimated using simulations from the posterior distribution of the response from the final model, averaged over the first and second decade of the time period. The probability of an outbreak has increased across most of Brazil. The Amazonian barrier has almost completely been eroded and the South Brazil border has moved further south. Maps were produced in R using the geobr package [32,35] (https://ipeagit.github.io/geobr/).

## Discussion

We found that the expansion of the dengue transmission zone is associated with temperature suitability, connectivity within the Brazilian urban network and urbanisation, and that the odds of future outbreaks significantly increase after both the vector and the virus have been introduced. This study builds on previous literature that showed the expansion of dengue across Brazil [6,7,17,26,43] and has updated the geographical barriers to transmission. The most recent epidemiological bulletins have shown that this expansion has continued in 2021 into previously unaffected parts of Acre, Amazonas, and further south into Paraná and Santa Catarina [44], highlighting the importance of monitoring the erosion of these barriers. To our knowledge, this is the first epidemiological modelling study to use the REGIC's levels of influence and show that there is an increased odds of dengue outbreaks in cities that are highly connected within the Brazilian urban network. However, this increase is not linear; regional capitals are considered less connected than metropoles but we found that the increase in odds were higher in these cities. Further investigation is needed to understand whether this is

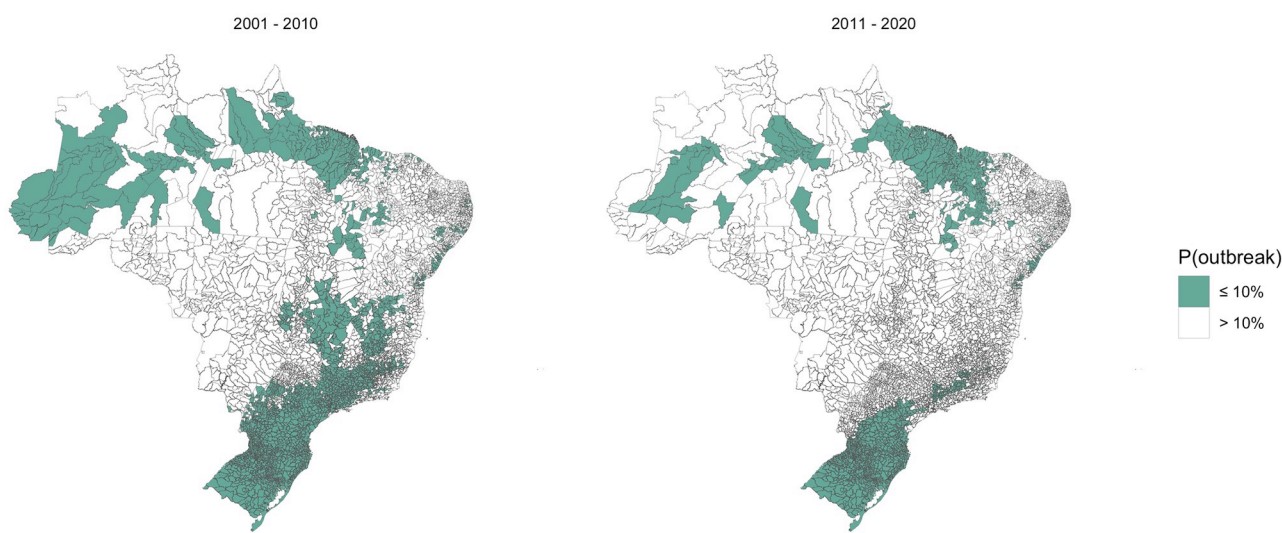

**Fig 11. Geographical barriers to dengue transmission in a) 2001–2010 and b) 2011–2020.** Maps showing areas where the probability of an outbreak was less than 10% on average in each decade of the 21st century. Between 2011–2020, only the 2 most southern states and the northern coast were fully protected from dengue transmission. Maps were produced in R using the geobr package [32,35] (https://ipeagit.github.io/geobr/).

related to human movement, as people more often travel to regional capitals from smaller cities than metropoles [29], or differences in socioeconomic factors and health-seeking behaviour that we were unable to detect at the municipality level.

Although this study focuses on Brazil, there is evidence that similar patterns are emerging in other parts of South America. In Argentina, previously protected cities in temperate regions are experiencing regular outbreaks, partially related to increasing temperatures but also as a result of human movement importing cases from other parts of the continent [23,24]. Rural parts of the Amazon, which were previously isolated from infected hosts and vectors, are also experiencing outbreaks, thought to be associated with increased connectivity between rural areas and larger cities [13,17]. The introduction of dengue into Acre in the Brazilian Amazon has been linked to increased connectivity across the state following the construction of a highway between the two largest cities, Rio Branco and Cruzeiro do Sul [26]. The impact of this connection can be observed in the data as the outbreak appears to jump from Rio Branco in the south of Acre to Cruzeiro do Sul in the north in 2014 rather than spreading to neighbouring regions which appears to be the case in the South (Fig 5). The introduction of dengue into the Amazon is particularly worrying as it is the ideal environment for the virus to thrive: lower than average access to basic services such piped water and refuse collection, and the ideal climate conditions for large epidemics [17,45].

Although this study extends our understanding of the expansion of the dengue transmission zone in Brazil, there are several limitations. Dengue case data used in this study was taken from Brazil's passive surveillance system, which has been found to differ in accuracy between regions, and between epidemic and non-epidemic periods [31]. To reduce the impact of reporting bias in our model, we used an outbreak indicator rather than case data as a response variable. The outbreak indicator used was chosen as it reflects the Brazilian Ministry of Health's definition [37]. However, the threshold of an outbreak is likely to differ across the country. In regions that historically experienced little or no transmission, even a small number of cases may be viewed as an outbreak. The choice of such a high threshold is likely to produce more conservative estimates of the transmission zone. When our results were compared to a lower outbreak threshold of 100 cases per 100,000 residents, we found the model conclusions

were consistent with the higher threshold. We found that both models using a fixed threshold outperformed the model based on the 75th percentile based on the area under the ROC curve (S7 Fig) and the Brier score. The model failed to pick up some of the temporal trends in the data, which may be a result of using stationary indicators of urbanisation and connectivity measured every 10 years. Information collected at a finer temporal scale may provide more insights into the impact of sudden expansions such as the effect of improved infrastructure in the Amazon [26].

Our model used the level of influence extracted from the REGIC studies [28,29] to account for the level of connectivity between cities within Brazil as a proxy for human movement. However, this indicator may simplify the process and miss important patterns. The hierarchical model assumed by REGIC assumes each small city is linked to a higher-level urban centre, such as the regional capitals and metropoles. It is evident that large and warm cities may propagate epidemic waves and maintain dengue transmission in their hinterland, while temperate metropoles in the South (Porto Alegre, Curitiba and São Paulo) do not play a relevant role in dengue diffusion in their region. Previous studies have found that imported cases driven by human movement are responsible for dengue outbreaks in temperate cities [24,25]. The choice of spatial connectivity assumption and data can lead to very different results and the use of the REGIC levels of influence as a spatial covariate rather than including the direct links may miss some important patterns [27]. Future work will aim to incorporate the complex urban network from the REGIC studies into a statistical framework to account for direct and indirect links between metropoles and regional capitals, and smaller urban centres in their hinterland.

Despite these limitations, we have shown that the expansion of the dengue transmission zone has continued into the 21st century, driven by increased temperature suitability in the South, a network of highly connected cities, and high levels of urbanisation. The introduction of dengue outbreaks into an area more than doubles the odds of future outbreaks, which is particularly concerning given the expansion has continued into 2021. Given the dynamic nature of the growing dengue burden, the barriers identified here will be outdated very quickly. We have highlighted the importance of focusing control strategies in areas at risk of future outbreaks as well as those within the established dengue transmission zone.

## Supporting information

**S1 Text. Supplementary material.** Additional information about the methods and materials used in this study and results of sensitivity analyses.
(DOCX)

**S1 Alternative Language Abstract. Translation of the Abstract into Portuguese by Rafael de Castro Catão.**
(DOCX)

**S1 Fig. The organisation of Brazil into a) 5 geo-political regions, and b) 27 federal units.**
Abbreviations: AC = Acre, AL = Alagoas, AP = Amapá, AM = Amazonas, BA = Bahia, CE = Ceará, DF = Distrito Federal, ES = Espírito Santo, GO = Goiás, MA = Maranhão, MT = Mato Grosso, MS = Mato Grosso do Sul, MG = Minas Gerais, PA = Pará, PB = Paraíba, PR = Paraná, PR = Pernambuco, PI = Piauí, RJ = Rio de Janeiro, RN = Rio Grande do Norte, RS = Rio Grande do Sul, RO = Rondônia, RR = Roraima, SC = Santa Catarina, SP = São Paulo, SE = Sergipe, TO = Tocantins. Maps were produced in R using the geobr package [32,35] (https://ipeagit.github.io/geobr/).
(TIF)

**S2 Fig. Average monthly mean temperature (˚C) in each Brazilian state January 2001—December 2020.**
(TIF)

**S3 Fig. The average number of months suitable for dengue transmission per year a) 2001–2010, and b) 2011–2020.** The average number of months with mean temperature between 16.2˚C and 34.5˚C aggregated to the two decades of data. Most of Brazil experiences suitable temperatures year-round apart from areas of South Brazil and areas of high altitude in the Southeast which experience cool winters. Maps were produced in R using the geobr package [32,35] (https://ipeagit.github.io/geobr/).
(TIF)

**S4 Fig. The percentage of residents living in urban areas of each municipality from the 2000 (a) and 2010 (b) censuses.** Levels of urbanisation differ greatly across Brazil, with the majority of Southeast and South Brazil living in urban areas in comparison to the North and Northeast which has a larger rural population. Maps were produced in R using the geobr package [32,35] (https://ipeagit.github.io/geobr/).
(TIF)

**S5 Fig. The proportion of municipalities in each region of Brazil experiencing an outbreak per year 2001–2020.** The proportion of municipalities affected by outbreak has increased since 2010 in every region of the country, although outbreaks in South Brazil are still focused on a small part of the region. Maps were produced in R using the geobr package [32,35] (https://ipeagit.github.io/geobr/).
(TIF)

**S6 Fig. Raincloud plots exploring the relationship between REGIC level of influence and a) urbanisation, b) access to piped water, and c) refuse collection.** Metropoles and regional capitals have higher levels of urbanisation and access to basic services than municipalities that had lower levels of connectivity within the urban network. Local centres were more varied in terms of basic services and urban levels than the other levels and covered a wide range of city types.
(TIF)

**S7 Fig. Receiver operating characteristic (ROC) curve for the final model (solid black line), the model using an outbreak threshold of over 100 cases per 100,000 residents (red dashed line), and the model using an outbreak threshold of over the 75th percentile (blue dashed line), compared to chance (black dashed line).** The closer to the top-left corner, the better the predictive ability of a model. As the ROC curve lies above the dashed reference line, this model performs better than chance.
(TIF)

**S8 Fig. The mean and 95% credible interval of the posterior distribution for each model covariate under different outbreak threshold definitions.** Coefficient estimates using the outbreak indicator based on the 75th percentile were noticeably smaller than the fixed threshold alternatives. The fixed threshold models (where outbreaks were defined as a dengue incidence rate of over 100 or 300) produced similar estimates, however the odds of an outbreak in municipalities after a previous outbreak was higher for the DIR = 100 model.
(TIF)

**S9 Fig. The probability of an outbreak estimated from the model for each year 2001–2020.** The mean probability of an outbreak estimated by taking 1000 simulations from the posterior

distribution of the response and transforming the outcome using a probit function. Maps were produced in R using the geobr package [32,35] (https://ipeagit.github.io/geobr/).
(TIF)

**S10 Fig. Comparison of different risk thresholds to define current geographical barriers to dengue outbreaks.** Municipalities were considered 'protected' if the probability of an outbreak was less than or equal to the threshold a) 0%, b) 5%, c) 10% or d) 15%. The threshold of 10% was chosen as it was the most comparable with previous studies. Maps were produced in R using the geobr package [32,35] (https://ipeagit.github.io/geobr/).
(TIF)

**S1 Table. Distribution of municipalities at each level of influence in the urban network, 2007 [28] and 2018 [29].** The number of municipalities classified as metropoles (largest cities in Brazil, connected throughout the entire country), regional capitals (large cities connected regionally and to metropoles), sub-regional capitals (cities connected locally and to the three largest metropoles), zone centres (smaller cities generally connected only to their neighbours), and local centres (smallest cities typically disconnected from the urban network).
(DOCX)

**S2 Table. Posterior mean and 95% credible interval (CI) estimates for linear effect parameters, shown on the adjusted odds ratio (aOR) scale, for alternative model formulations.** Coefficient estimates for models assuming an outbreak threshold of over 100 cases per 100,000 (medium risk model), an outbreak threshold of over the 75th percentile of incidence rates, and using temperature suitability for *Aedes aegypti* only.
(DOCX)

**S3 Table. Model comparison statistics.** Area under the receiver operator curve and Brier scores for models assuming an outbreak threshold of over 300 cases per 100,000 residents (high risk model), over 100 cases per 100,000 (medium risk model), over the 75th percentile of incidence rates, and a model including the number of months considered extremely wet.
(DOCX)

## Author Contributions

**Conceptualization:** Sophie A. Lee, Rachel Lowe.

**Data curation:** Sophie A. Lee.

**Formal analysis:** Sophie A. Lee.

**Investigation:** Sophie A. Lee, Rafael de Castro Catão, Christovam Barcellos, Rachel Lowe.

**Methodology:** Sophie A. Lee, Theodoros Economou, Rachel Lowe.

**Software:** Sophie A. Lee, Theodoros Economou, Rachel Lowe.

**Supervision:** Theodoros Economou, Rachel Lowe.

**Validation:** Sophie A. Lee, Rachel Lowe.

**Visualization:** Sophie A. Lee.

**Writing – original draft:** Sophie A. Lee.

**Writing – review & editing:** Sophie A. Lee, Theodoros Economou, Rafael de Castro Catão, Christovam Barcellos, Rachel Lowe.

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
