## [Decision Letter · Decision Letter 0]

18 Oct 2021

Dear Ms Lee,

Thank you very much for submitting your manuscript "The impact of climate suitability, urbanisation, and connectivity on the expansion of dengue in 21st century Brazil" for consideration at PLOS Neglected Tropical Diseases. As with all papers reviewed by the journal, your manuscript was reviewed by members of the editorial board and by several independent reviewers. The reviewers appreciated the attention to an important topic. Based on the reviews, we are likely to accept this manuscript for publication, providing that you modify the manuscript according to the review recommendations. 

Sincerely,

Hannah E. Clapham

Associate Editor

Paul Newton

Deputy Editor

Reviewer's Responses to Questions

**Key Review Criteria Required for Acceptance?**

**Methods**

-Are the objectives of the study clearly articulated with a clear testable hypothesis stated?

-Is the study design appropriate to address the stated objectives?

-Is the population clearly described and appropriate for the hypothesis being tested?

-Is the sample size sufficient to ensure adequate power to address the hypothesis being tested?

-Were correct statistical analysis used to support conclusions?

-Are there concerns about ethical or regulatory requirements being met?

Reviewer #1: (No Response)

Reviewer #2: The authors only explored one environmental factor, which is temperature suitability, for explaining the expansion of dengue. Other climatic factors such as rainfall and humidity may also impact the Aedes aegypti vector population. Please explain the rationale behind choosing temperature as the only environmental factor here. 

The study created a binary outbreak indicator based on case counts in the dengue surveillance system in Brazil as the response variable of the GAM model. Please comment on how the surveillance system is sensitive to change in health seeking behavior or access to healthcare over time, and how it may impact the model output.

**Results**

-Does the analysis presented match the analysis plan?

-Are the results clearly and completely presented?

-Are the figures (Tables, Images) of sufficient quality for clarity?

Reviewer #1: (No Response)

Reviewer #2: The results are clear overall. One minor comment is that under "model result" on line 309, the authors stated that the increased OR suggests that dengue becomes established once the virus is introduced. It would be helpful to present dynamics of the four circulating serotypes otherwise the statement may be a stretch.

**Conclusions**

-Are the conclusions supported by the data presented?

-Are the limitations of analysis clearly described?

-Do the authors discuss how these data can be helpful to advance our understanding of the topic under study?

-Is public health relevance addressed?

Reviewer #1: (No Response)

Reviewer #2: The conclusions are clear and well presented.

**Editorial and Data Presentation Modifications?**

Reviewer #1: (No Response)

Reviewer #2: (No Response)

**Summary and General Comments**

Reviewer #1: Review PNTD The impact of climate suitability, urbanisation, and connectivity on the expansion of dengue in 21st century Brazil 

This study aims at exploring the changes in transmission in Brazilian regions. The goal is to identify the new limits to dengue presence in Brazilian regions and exploring the drivers of dengue expansion using a statistical model. This is a well-written, interesting study with clear assumptions and methodology. I was not familiar with the literature on the association of human transportation networks and dengue spread in Brazil, but the background and approach used were explained very clearly, although I would have liked to understand more what caused differences between urban centers. The analyses and figures are very informative. My only concern is with the choice of threshold as the response variable in the model. I would like to have a better understanding of the choice of incidence chosen, and I would like to see a better comparison between models with different threshold choices. Could we see a sensitivity analysis exploring the impact of 300 cases per 100,000 with other thresholds’ choices (20, 50, etc.) and formal model comparisons? Apart from this I only have a few suggestions. 

A few minor points:

l.126-7. Reference for rank of country by human population size needed.

l.140: I could not find the code associated with this link.

l.254-256: Not sure I understand the choice of threshold explored in this analysis. I would appreciate if this could this be specified in the methodology section and in the figure S9 legend. What is the impact of a lower threshold 30, 50 per 100,00?

l.273: There are many supplementary figures references. It would be helpful to order the plots in the Supplementary material in the order they appear in the text, e.g., here S8 could be ranked S5 and so on.

l.279: I’m not convinced figure 5 is useful to support this statement. I suggest either removing the sentence and figure or refer to table 1/figure 6 as they present strong evidence of the impact of previous outbreaks.

l.306-307: As I said before, my only reserve for this part of the analysis is that it is dependent on the quantitative choice of threshold. I would be interested in seeing a formal quantitative analysis of the relationship between incidence threshold and the likelihood of having more outbreak of similar strength (or at least seeing better support for this choice). 

l.313-314: I’m not sure there is a strong enough evidence of a higher risk of outbreak after an outbreak year. The risk seems to be lower compared to the model with an outbreak any previous year. Could it be outbreaks that overlap over two calendar years instead of a higher risk of outbreak after an outbreak year? A cautious reformulation of this sentence or better explaining the limits of this analysis could help.

l. 340-349: Could it be that other confounders exists that may affect dengue reporting rates? E.g., regional centers having a higher number of doctors per capita than metropoles? 

l.351-352. I agree that the models reach to the same conclusions, but I do see differences in the parameter estimates where the estimates for some models were not within the bounds of the main model. Also, I am curious what the ROC figure (S9) looks like for the other model formulations (100 per 100,000 incidence, model with each covariate), could they be plotted in dashed or grey-ish line?

l. 354-358: Very informative part of the analysis! I really appreciate the information and the visualization of the spatially heterogeneous risk of dengue outbreak. 

l. 375: In some instances, “climate suitability” is used but temperature suitability may be more appropriate as the temperature is the only variable used for this parameter estimate (and the words "temperature suitability" are used in the methods section). 

l.407-409: Methods for this analysis? Since the figures for this analysis are presented in the main text the methods, could be found in the methods section of the main text?

Reviewer #2: Study by Lee et al. built upon previous literature that showed geographic expansion of dengue transmission in Brazil, and it updated the changing geographic barrier to dengue transmission in Brazil. It explored several drivers behind such expansion and provided additional evidence on the impact of temperature suitability and urbanization on dengue outbreaks in Brazil. The manuscript is well-written and has important policy implications.

PLOS authors have the option to publish the peer review history of their article (what does this mean?). If published, this will include your full peer review and any attached files.

Reviewer #1: No

Reviewer #2: No

Figure Files:

Data Requirements:

Reproducibility:

References

---

## [Editor Report · Decision Letter 1]

24 Nov 2021

Dear Ms Lee,

We are pleased to inform you that your manuscript 'The impact of climate suitability, urbanisation, and connectivity on the expansion of dengue in 21st century Brazil' has been provisionally accepted for publication in PLOS Neglected Tropical Diseases.

Best regards,

Hannah E. Clapham

Associate Editor

Paul Newton

Deputy Editor

---

## [Editor Report · Acceptance letter]

6 Dec 2021

Dear Ms Lee,

We are delighted to inform you that your manuscript, "The impact of climate suitability, urbanisation, and connectivity on the expansion of dengue in 21st century Brazil," has been formally accepted for publication in PLOS Neglected Tropical Diseases.

Best regards,

Shaden Kamhawi

co-Editor-in-Chief

Paul Brindley

co-Editor-in-Chief
